# On the Effect of Training Convolution Neural Network for Millimeter-Wave Radar-Based Hand Gesture Recognition

**DOI:** 10.3390/s21010259

**Published:** 2021-01-02

**Authors:** Kang Zhang, Shengchang Lan, Guiyuan Zhang

**Affiliations:** Department of Microwave Engineering, Harbin Institute of Technology, Harbin 150001, China; zhangkang@hit.edu.cn (K.Z.); 19S005011@stu.hit.edu.cn (G.Z.)

**Keywords:** hand gesture recognition, FMCW radar, Fine tune, millimeter-wave band

## Abstract

The purpose of this paper was to investigate the effect of a training state-of-the-art convolution neural network (CNN) for millimeter-wave radar-based hand gesture recognition (MR-HGR). Focusing on the small training dataset problem in MR-HGR, this paper first proposed to transfer the knowledge with the CNN models in computer vision to MR-HGR by fine-tuning the models with radar data samples. Meanwhile, for the different data modality in MR-HGR, a parameterized representation of temporal space-velocity (TSV) spectrogram was proposed as an integrated data modality of the time-evolving hand gesture features in the radar echo signals. The TSV spectrograms representing six common gestures in human–computer interaction (HCI) from nine volunteers were used as the data samples in the experiment. The evaluated models included ResNet with 50, 101, and 152 layers, DenseNet with 121, 161 and 169 layers, as well as light-weight MobileNet V2 and ShuffleNet V2, mostly proposed by many latest publications. In the experiment, not only self-testing (ST), but also more persuasive cross-testing (CT), were implemented to evaluate whether the fine-tuned models generalize to the radar data samples. The CT results show that the best fine-tuned models can reach to an average accuracy higher than 93% with a comparable ST average accuracy almost 100%. Moreover, in order to alleviate the problem caused by private gesture habits, an auxiliary test was performed by augmenting four shots of the gestures with the heaviest misclassifications into the training set. This enriching test is similar with the scenario that a tablet reacts to a new user. The results of two different volunteer in the enriching test shows that the average accuracy of the enriched gesture can be improved from 55.59% and 65.58% to 90.66% and 95.95% respectively. Compared with some baseline work in MR-HGR, the investigation by this paper can be beneficial in promoting MR-HGR in future industry applications and consumer electronic design.

## 1. Introduction

Hand gesture recognition (HGR) has developed prosperously in recent years, allowing various contacted and contactless solutions for different industrial and daily life backgrounds. Both Electromyogram (EMG) and Mechanomyogram (MMG) have been explored based on the human anatomy to acquire muscle contractions or joint movements [1,2]. Although these two contacted sensors can be wearable to any location without range constraints, they are still limited by the fact that wire connections between the electrodes and circuits may heavily reduce user comfort. As the most popular contactless solution, vision-based hand gesture recognition utilizes images or videos of gestures captured by cameras with a fair cost and mature interface with consumer electronics. However, this solution is imperfect due to high dependency on illumination and potential privacy leakage. Inspired by the rapid development in millimeter-wave radar sensors, electromagnetic (EM) waves are an emerging alternative in HGR for the benefits of non-illumination-dependence and non-contact sensing ability. Many publications have introduced the milestone work of Soli project, using a 60 GHz Frequency Modulated Continuous Wave (FMCW) millimeter-wave radar sensor to control a wearable watch [3,4]. Furthermore, the authors of [5] proposed a contactless strategy of assisting car drivers with a millimeter-wave radar and computer vision. Inspired by the development of deep learning, millimeter-wave radar-based hand gesture recognition (MR-HGR) has launched many new explorations in human–computer interaction (HCI) for future industrial applications and consumer electronics design. The authors of [4,6] proposed an optimized network structure by using an end-to-end convolution neural network (CNN) and long short-term memory (LSTM) to process the time-varying range-Doppler map (RDM) sequences. In addition, the authors of [7] employed a recurrent three-dimensional CNN with a Connectionist Temporal Classification (CTC) algorithm to perform classification of dynamic hand gestures. In this case, numerous state-of-the-art methods for MR-HGR follow a generic workflow by using deep learning to process radar echo signals, extracting the signatures, and implementing the recognition in [8,9,10]. Nevertheless, unlike visual images, radar measurements are not inherently images, but actually a time-stream of complex In-phase and Quadrature (I/Q) data from which line-of-sight distance and radial velocity can be computed [11]. However, full training of any neural network for MR-HGR is far different from training in computer vision. First, all the training datasets used in the previous works for MR-HGR are smaller than the number of network parameters required for training. Inadequate training dataset may significantly reduce the network’s ability to generalize, even resulting in further overfitting and convergence problems. Second, it is also difficult to construct a labeled dataset with a comparable size to ImageNet in MR-HGR due to different radar sensor hardware specifications such as frequency band, modulation bandwidth, and modulation waveform. Additionally, hand gestures are private because personal habits may have a significant impact on the micro-Doppler signatures in the radar echo signals. Therefore, direct immigration of many state-of-the-art deep learning models from computer vision to MR-HGR is not applicable. The key solutions to the problems should be explored in both signal processing methods and deep learning strategies. An RGB-like visual representation of hand gestures is requisite for applying CNN in MR-HGR. Moreover, borrowing the experiences how CNN learns novel classes using only a few examples in computer vision can contribute to the training with limited data samples in MR-HGR.

Therefore, this paper primarily focused on the solution of MR-HGR training problems by fine-tuning state-of-the-art CNN models in computer vision for MR-HGR and the performance evaluation of these fine-tuned networks. In Section 2, we proposed a parameterized representation of dynamic hand gestures based on temporal space-velocity (TSV) spectrograms to represent six categories of common gestures used in HCI and constructed the datasets with TSV spectrograms from nine volunteers. A framework of the transfer learning strategy based on the data samples from a 77 GHz FMCW radar sensor was firstly presented in Section 3. Section 4 discussed the evaluation experiments and gave further result analysis. Finally, we concluded the paper in Section 5.

## 2. Parameterized Representation of Dynamic Hand Gestures

A hand gesture is physically interpreted as the multiple freedom joint dynamics of the fingers, palm, and wrist, with correspondingly different velocity, trajectory angles, and scattering energy. Due to the rapid development of circuit and antenna design in millimeter-wave band, MR-HGR can provide a highly integrated solution of HCI for many consumer electronics and industry applications. Furthermore, the development of deep learning has promoted the feasibility of MR-HGR by intelligently processing the radar echo signals and accurately categorizing different gestures. Due to the excellent fitting ability, deep learning is extensively used for computer vision and natural language processing tasks. However, the choice of deep learning models in MR-HGR such as CNN, LSTM, or CNN plus LSTM is highly relevant to the radar sensor and processing procedures of radar echo signals. A typical signal processing chain for radar echo signals are range–time, Doppler–time, and range–Doppler patterns for the target gestures in front of the radar. Therefore, LSTM is better at processing time-related image sequences, while CNN outperforms most state-of-the-art machine learning algorithms in image recognition. If a hand gesture can be recorded in terms of an image rather than an image sequence, CNN can be an effective solution. As one of the most powerful deep learning algorithm, CNN can take in an input image, assign learnable weights and biases to various targets on the image, and differentiate one from the others. A CNN generally consists of an input layer, an output or classification layer, and multiple hidden layers. Many classic CNN models such as ResNet [12] and DenseNet [13] integrate a convolution, pooling, and full-connection or softmax layer as a hidden layer to achieve a high accuracy rate on ImageNet. Meanwhile, in order to run CNN on computationally constrained devices, many publications have investigated lightweight networks and balance the speed–accuracy trade-off, including Xception [14], MobileNet [15], MobileNet V2 [16], ShuffleNet [17], and ShuffleNet V2 [18]. No matter which model is used for MR-HGR, the first challenge is how to train these models with the limited training data efficiently. Therefore, the following part gives a close look at the dataset and the signatures that can be extracted from the dataset.

### 2.1. Dataset Collection

A total of six gestures are shown in Figure 1, namely, “page-open”, “page-close”, “page-up”, “page-down”, “page-zoom-in”, and “page-zoom-out”. The aforementioned six gestures were selected as the basic touch screen gestures to interact with many tablets such as IPad and other consumer electronic products, and thus urgently call for close attentions in MR-HGR. From the view of radar, these gestures are different in velocity, trajectory angles, and scattering signatures. For example, “page-up” and “page-down” in Figure 1 can be interpreted as the pairwise motions with fingers, the palm and the wrist moving radially towards the radar at the same velocity. Unlikely, “page-open” and “page-close” can be interpreted as the trajectories of finger and palm joints with the invariant wrist. More precisely, gestures such as “page-zoom-in” and “page-zoom-out” can be interpreted as only the trajectories of thumb and index fingers with shorter duration and smaller radar cross section. Four volunteers proficient in using a tablet were invited to collect a total of 3504 gesture examples as the local datasets, named as shown in Table 1. In order to enhance the data diversity, four volunteers were selected within different ages, genders, and heights.

### 2.2. Temporal Space-Velocity Spectrograms

As discussed above, the form of input data has a key impact on the choice of the deep learning algorithms. The majority of the state-of-the-art researches directly select the RDM sequence as the input data and correspondingly utilize LSTM or Recurrent Neural Network (RNN) for sequence prediction problems. This combination also entails the increase of the network parameters and overfitting risks, especially without adequate training data. In this case, many signal processing strategies have been proposed for the new input form such as I/Q plot or temporal evolution of micro-Doppler signatures to reduce the network complexity [5,19]. However, I/Q plot may be imbalanced due to the radar sensor hardware cracks. Pure micro-Doppler signatures discard the target’s range patterns which represent hand position and pose in front of the radar. Hereby, more work is desired for better representation of dynamic hand gestures and the following part in this subsection presents a parameterized representation of dynamic hand gestures.

In the radar signal processing, a FMCW radar can identify the target by transmitting a sequence of chirp signals in the regular cycles of $N_{f}$ frames and then split the radar echo signals to construct a sequence of RDMs. Each RDM is a matrix of $N_{r}\times N_{d}$ cells with respect to a frame of chirps, where $N_{r}$ and $N_{d}$, respectively, denote the indices of cells along the range and Doppler axes. The size of each evenly divided cell is Δr×ΔD, where Δr and ΔD, respectively, represent the resolutions of range and Doppler measurements. The intensity of each cell $m_{n}\left(n_{r},n_{d}\right)$ on the RDM in the *n*th frame is defined as $m_{n}\left(n_{r},n_{d}\right)$.

The instant existence of a point-like target at range *R* with a moving velocity *v* towards radar is determined by the intensity of the cell $m_{n}\left(n_{\hat{r}},n_{\hat{d}}\right)$:(1)$$\left\{\begin{array}{c} n_{\hat{r}}=\left(\frac{2f_{c}v}{c}+\frac{2BR}{T_{c}c}\right)\frac{N_{r}}{f_{s}}\\ n_{\hat{d}}=\frac{2vf_{c}T_{sl}N_{d}}{c} \end{array}\right.$$
where $T_{sl}$ is the time interval between two adjacent chirps, $f_{c}$ is the carrier frequency, *B* is the bandwidth, $T_{c}$ is the chirp duration, $f_{s}$ is the baseband sample frequency, and *c* is the light speed. For a multiple point-like target scenario, the target can be represented by a set of cells, usually a few peaks on the RDM.

MR-HGR is generally performed close to the millimeter-wave radar sensor, i.e., less than 1 meter. Therefore, the reflection energy from the hand is so strong that the noise and scattering energy from other objects in the scenario can be ignored. A 4 GHz modulated frequency bandwidth can provide a centimeter level of Δr according to Equation (Equation 1). Therefore, a human hand always occupies several adjacent cells on the RDM instead of a single cell. Meanwhile, a complete hand gesture has different velocity characteristics from the start to the end due to different extents of the flexion/extension, abduction/adduction, and rotation of the joints. Thus, velocity evolution can represent the hand gesture difference. However, it is still infeasible to use raw RDMs as the network input because it increases the computation and resource cost greatly. To alleviate the drawbacks of using raw RDMs directly as the network input, and help the model to focus attention on gesture recognition, we proposed a TSV parameterized representation of hand gestures and explained how to derive this representation from the RDMs in the following part.

As stated above, the cells representing a static hand is within the proximity of $n_{\hat{d}}=0$. If the hand moves, the interest region on the RDM moves to the cells within the proximity of $n_{\hat{d}}=\frac{2vf_{c}T_{sl}N_{d}}{c}$. That is, only the cells in the updated interest region are worthy of being traced in MR-HGR.

In addition, the energy regarding to each frame firstly increases when the movement starts and decreases after it reaches the maximum until the hand becomes static. The entire energy variation is shown in Figure 2. Therefore, it enables the possibility of using a sliding filter to extract the frames representing the hand movements from all $N_{f}$ frames. Equation (Equation 2) denotes the center of the extracted frames based on the maximum energy of all the frames by the sliding filter. Centered at $W_{c}$, a 2L+1 length of frames can be extracted from the frame sequence. *L* denotes the sliding filter length determined by the time duration of a typical hand movement, mostly less than than 0.4 s.
(2)$$W_{c}\underset{n}{=\arg\max}\;\;\left[\sum_{n_{r}=1}^{N_{r}}\sum_{n_{d}=1}^{N_{d}}∥m_{n}\left(n_{r},n_{d}\right)∥\right]$$

With the knowledge of the extracted frames, the next step is to extract the cells in the interest region. The extraction is performed by averaging $n_{r}$ with respect to the peak on the RDM in each extracted frame, as shown in Equation (Equation 3).
(3)$$\hat{R}=\left\lfloor \frac{1}{2L+1}\sum_{n=W_{c}-L}^{W_{c}+L}[\underset{n_{r}}{\arg\max}\;\;\sum_{n_{d}=1}^{N_{d}}∥m_{n}\left(n_{r},n_{d}\right)∥]\right\rfloor$$

Considering the relationship between Δr and hand size, the cells adjacent to $\hat{R}$ can represent the hand. In this case, velocity evolution at $\hat{R}$, $\hat{R}-1$, and $\hat{R}+1$ from $\left(W_{c}-L\right)$th to $\left(W_{c}+L\right)$th frames could disclose the difference of hand gestures. Short-time frequency spectrum analysis in the Doppler domain could derive the velocity evolution at $\hat{R}$ and adjacent ranges $\hat{R}-1$, and $\hat{R}+1$, respectively. Each processed short-time frequency spectrum can be used as a channel of an RGB-like image. For instance, it helps to understand the definition of TSV spectrogram if we build a three-dimensional image with the spectrums corresponding to $\hat{R}-1$, $\hat{R}$, and $\hat{R}+1$, like using the spectrums to represent the red, green, and blue colors in RGB models. Consequently, a RDM sequence can be greatly compressed into the parameterized representation in terms of an RGB-like image, namely, TSV spectrogram. The process is detailed in Figure 3 and the TSV spectrogram can be visually displayed at the bottom right in this figure.

As millimeter-wave radar is sensitive to velocity information, different hand sizes and habits of each volunteer produce inconsistency in the TSV spectrograms. Figure 4 shows two TSV spectrograms representing the “page-zoom-out” gestures of two volunteers. Compared with VolA, VolB shows a wider Doppler distribution in the frequency domain and stronger power of the radar echo signals. This dissimilarity can be interpreted by the different motions of two volunteers performing the same gesture, more specifically by the different separating velocities of the thumb and the index finger in the “page-zoom out”. The TSV spectrograms are also affected by human anatomy information such as different palm sizes and finger lengths of the two volunteers. Training a network with the data samples of VolA may not lead to good test result with data samples of VolB. As opposed to computer vision, the private habits of data samples increases the training complexity in MR-HGR.

## 3. Transfer Learning for Mr-Hgr

Transfer learning is a process to take a model that has already been trained for a given task and make it perform another similar task. Current popular CNN models including AlexNet [20], GoogLeNet [21], and VggNet [22] have been extensively investigated with their implementations and pretrained weights on a standard dataset such as ImageNet. Inspired by knowledge transferring in computer vision, our baseline model follows the standard transfer learning procedures in [23]: pretraining the selected network and fine-tuning.

### 3.1. Pretraining and Fine-Tuning

Here, transfer learning usually refers to the ability of a system to recognize and apply knowledge and skills learned in previous domains/tasks to novel domains/tasks. The overall procedure is illustrated in Figure 5. Under such a circumstance, radar echo signals from a 77 GHz radar sensor were processed to set up the local MR-HGR dataset. The parameters of the pretrained CNN model learned to be optimal from the local MR-HGR dataset.

In the pretraining stage, our purpose was to construct a network with a signature extractor $f_{\theta }$ and a classifier $C(\cdot |W_{b})$ from scratch by minimizing a standard cross-entropy classification loss $L_{pred}$ using ImageNet [24], namely, $x_{i}\in X_{b}$. All the pretrained models used in this paper were models built in Pytorch and typically trained on one-thousand classes of ImageNet, and once fine-tuning of the models was completed, we replaced the final layer of the model as described in Section 3.2. Normally, during the fine-tuning phase, we can either fix θ or involve it in training. In this paper, we did not fix the pretrained network parameter θ in our signature extractor $f_{\theta }$ and train the $f_{\theta }$ with a customized classifier $C(\cdot |W_{n})$ by minimizing a standard cross-entropy classification loss $L_{pred}$ using the training samples in the MR-HGR, namely, $x_{i}\in X_{n}$.

### 3.2. Customization of State-of-the-Art CNN Models for Fine-Tuning

#### 3.2.1. ResNet

ResNet [25] is a CNN model proposed by Microsoft Research. In 2015, the network featured easy optimization and the ability to improve accuracy by increasing the depth by a considerable amount. Its internal residual blocks use jump connections to alleviate the problem of gradient disappearance caused by increasing the depth in deep neural networks. Later in 2016, the authors of [12] updated ResNet using identity mapping to achieve a higher accuracy on ImageNet. Inspired by the work in [12], the same model pretrained on ImageNet was used in our evaluation except a customized 6-way classifier used to replace the last layer of the original network.

#### 3.2.2. DenseNet

DenseNet [13] shares the same basic idea as ResNet, but it constructs dense connections between all previous and subsequent layers. In comparison to ResNet [12], DenseNet not only alleviates the problem of gradient vanishing, but also greatly reduces the number of parameters [13]. For our MR-HGR task, DenseNet models with 121, 161, and 169 layers, as described in [13], were used. The DenseNet models were all pretrained on ImageNet and our customized full connection layer with softmax layer replaced the original top layer.

#### 3.2.3. MobileNet

MobileNet V1 [15] was released in 2017 to efficiently maximize model accuracy for a variety of use cases under limited computing resources. An upgraded version, MobileNet V2, was released as the next-generation lightweight network [16]. To fine-tune MobileNet V2 on MR-HGR dataset, we used a 6-way classifier to truncate the original softmax layer with the model weights pretrained on ImageNet.

#### 3.2.4. ShuffleNet

ShuffleNet [17] is a lightweight model proposed in 2018 as an improved version of ResNet by using group convolution and channel shuffle. Later, ShuffleNet V2 was proposed as an upgraded version with higher accuracy than ShuffleNet V1 and MobileNet V2 [16]. Likewise, fine-tuning the pretrained MobileNet V2 in MR-HGR was done by truncating the original softmax layer and replacing it with our own.

## 4. Experiment Results and Discussions

In order to evaluate the proposed methods for MR-HGR, a number of experiments were implemented with a 77 GHz radar sensor. In the experiments, whether CNN and TSV can work well with radar data samples was anticipated for further verification. In addition, a series of comprehensive cross tests were performed to show the response of the fine-tuned networks to the data samples that have never been learned previously and how to alleviate the negative effects of private hand gesture habits in MR-HGR. In the experiment, our main focus was to fine-tune the milestone classical models ResNet with 18, 50, and 152 layers and DenseNet with 121, 161, and 169 layers, as well as the lightweight models MobileNet V2 and ShuffleNet V2 published in recent years.

### 4.1. Hardware Setup

The radar sensor used in our experiments was Texas Instruments AWR1642boost radar set. This radar sensor operates at 77–81 GHz. The horizontal detection angle of the antenna is 120 degrees, and the vertical detection angle is 60 degrees. We selected the radar parameter configuration with a modulated frequency bandwidth of 3.2 GHz, baseband sampling rate of 2.4 MHz, ramp time of 33 s, and the frame length of 32 ms. This parameter achieved a range resolution of 4.62 cm and a velocity resolution of 6.75 cm/s. Local datasets shown in Table 1 were constructed based on this radar sensor, and nine volunteers were requested to sit on a chair with right hand facing the radar sensor at a distance of roughly 0.4 m and to complete gestures shown in Figure 1 in their own way. A desktop computer was used in the experiments with a single Graphics Processing Unit (GPU), GTX 1080, the memory of 32 GB, processor clock 33 MHz and the Windows 10 64 bits system. Software used included Python, Pytorch, Cuda, and CuDNN library.

### 4.2. Training Setup

In this work, radar echo signals require further processing to become visualized in the modality of a TSV spectrogram. This modality shows the temporal correspondence of range and Doppler rather than hand poses.

An Adam optimizer [26] was used for all experiments with the weight Decay 0 and a Nesterov momentum of 0.9, which uses momentum and adaptive learning rates to accelerate convergence. all the models were fine-tuned with a relatively small learning rate 0.001 and a batch size of 20 to avoid GPU memory outage. Each model was trained with 100 epochs and finally showed only the test accuracy at the 100th epoch.

No data augmentation was done in all the training process in the experiments. Although images can be arbitrarily rotated, flipped, or cropped, this cannot work well with radar sensing because any tiny change in the radar echo signals or on the processed spectrograms may make the potential features representing a totally different gesture.

To build up a baseline model, we first trained the models on the 75% of VolA dataset from scratch using random weights to initialize the model. Then the same training dataset was used in fine-tuning test with pretrained model. Therefore, there are totaling 900 training samples in the baseline and fine-tuning tests. The configuration of the baseline and fine-tuning tests is shown in Figure 6.

In both the baseline and fine-tuning tests, testing each model included two parts: self-testing (ST) and cross-testing (CT). The former test has been performed by the work in the state-of-the-art MR-HGR literature by using the data samples of the same volunteers for both the training and testing. The latter test strictly distinguishes the testing dataset from the training set by using the independent testing data samples from the volunteers never existing in the training set. Unlike ST, CT is helpful in getting an idea about how the fine-tuned models can generalize to a stranger’s hand gesture. The different configurations of the testing datasets in ST and CT are given in Figure 6.

### 4.3. Test Results and Analysis

A k-fold (k = 5) cross-validation procedure was implemented in each test to avoid the limitations and peculiarities of a fixed partitioned dataset. This utilization of the k-fold cross-validation has a more pronounced effect on the small-scale dataset, i.e., the datasets used in this paper. By reducing the variance associated with a single partition of the split training or testing dataset, k-fold cross-validation is anticipated to verify whether the model is overfitting in the real usage of MR-HGR. In this subsection, test accuracies in baseline test and fine-tuning test were compared to show whether the fine-tuned models can generalize to the radar date samples.

#### 4.3.1. Baseline Test

Both of ST and CT were performed in the baseline test without fine-tuning the networks. The role of ST in the baseline test is to reproduce the strategy used in much previous work in MR-HGR. Table 2 shows the test results in the baseline column. All the ST accuracies surpass 97%. However, CT accuracies fluctuate in a low range from 63% to 86%, which implies that the models may get greatly overfitted to the training set collector, VolA. Although test results in ST are impressive, more than 20% average accuracy declination in CT reveals that using state-of-the-art CNN models in computer vision directly without any concerns about radar dataset characteristics is not enough to promote the MR-HGR in the future applications.

#### 4.3.2. Fine-Tuning Test

Different from the baseline test, a fine-tuning test was then performed with pretrained weights and the customized CNN models. Both ST and CT were performed in the fine-tuning test. Table 2 also shows the fine-tuning test results in the fine-tuning column, and Figure 7 shows the CT confusion matrices regarding to each model after five folds using the radar data samples from three tested volunteers: VolB, VolC, and VolD.

It is not surprising that test accuracy in ST almost approaches to 100%. More importantly in CT, the highest test accuracy of 93.47% was achieved by ShuffleNet V2 after being fine-tuned. Following the light weight model, the classical model ResNet-18 achieved the second highest accuracy of 91.77%. All the models in CT after fine-tuning achieved higher test accuracy, at least 8.1% improvement compared with CT in the baseline test. A small dataset is enough to enable CNN models to transfer knowledge from computer visions to new radar domain by fine-tuning.

#### 4.3.3. Enriching Test

The confusion matrices in Figure 8 provide a thorough understanding of the misclassifications with respect to all the hand gestures regarding to each tested volunteer in CT. A total number of 128 × 6 × 3 = 2304 data samples were used to test the gestures of each volunteer: VolB, VolC, and VolD. It is still discovered that the misclassifications regularly occur in the specific gesture recognition such as VolB’s “page-close” and VolD’s “page-zoom-out”. About 33.3% “page-close” of VolB is misclassified as “page-down” and nearly 32.3% “page-zoom-out” of VolD is misclassified as “page-up”. The same heavy misclassification is not discovered when VolC’s gestures were tested and other gestures of VolB and VolD were tested. It is believed that the main reasons for these misclassifications are not the incapability of the fine-tuned models but possibly that private gesture habits play an important role in the misclassifications.

In order to further alleviate the misclassifications, an auxiliary test was performed by enriching the training dataset with four shots regarding to the heaviest misclassified gestures VolB’s “page-close” and VolD’s “page-zoom-out”, respectively. This enrichment of the training dataset also aims to emulate the scenario how to train a tablet against versatile user gesture habits. The empirical guidance of training a tablet or other electronic products to know its new user’s biosignatures is only based on small shots provided by the user rather than hundreds of repetition. Moreover, it is anticipated that the enrichment can enhance the models to learn private gesture habits.

As the representative models with highest average accuracies in the above test, ResNet-18 and ShuffleNet V2 were selected and fine-tuned with the enriched training datasets. After 5-fold cross-validation, the confusion matrices are shown in Figure 9. The training dataset for each fold is 75% VolA and 4 randomly chosen samples totaling 4 + 900 = 904 samples and the testing dataset for each fold has a total amount of 128 × 6 = 768 samples for each of VolB and VolD. As seen in Figure 9, the misclassification of VolB’s “page-close” as “page-down” is reduced from 32.71% to 9.06% and the misclassification of VolD’s “page-zoom-out” as “page-up” declines from 32.28% to 3.71%. Therefore, the misclassification reduction between Figure 8 and Figure 9 supports that misclassifications resulted from different private hand gesture habits can be alleviated by the proposed enrichment.

### 4.4. Discussion

Unlike the classification in computer vision, MR-HGR is not only the classification of the right category of the gesture, but also the classification of who has actually gestured due to the personal gesture habits. From the results shown in Section 4.3, it can be seen that deep learning can solve the problem of using millimeter-wave for contactless recognition of different gestures with respect to different people. The miniaturized design of millimeter-wave radar sensors also supports applications under versatile backgrounds such as smart watches, tablets, and mobile phones. Contactless HCI, in practice, generally needs to be deployed on mobile devices to serve hundreds of millions of users with different private habits. As stated above, it is very challenging in MR-HGR that a large amount of labeled data should be provided to make the CNN model generalized to the local dataset. In addition, training process requires extensive computations and correspondingly becomes time-consuming. Therefore, the concept of fine-tuning state-of-art CNN model has been introduced and demonstrated, as it facilitates boosting of accuracy and reducing acquisition time of available models with the limited radar data samples.

Compared with some baseline work in [4,5,10,19], some representative results in MR-HGR are summarized in Table 3. The six categories of gestures collected in this paper are the most commonly used gestures when operating devices such as tablet, and they are more representative than the gestures collected in [4,5,9], in which some gestures beyond our study are the simple repetitions of the gestures in our paper. For example, the authors of [9] studied “single blinking”, which is the same as “page-zoom-in” in our paper. The recognition of the repeated gestures such as “double blinking” can be implemented based on the recognition of two single gestures “single blinking”. Table 3 also shows that majority of the gesture recognition in previous work and our paper have high accuracy in ST, but only our paper and [9] performed CT. The method in our paper achieved 93.47% average accuracy. The CT accuracy in both our paper and [9] are smaller than these in ST, which is a side effect of the model’s accuracy higher in the private gestures that have been learned than in other people’s gestures that have not been learned by the model.

With the listed information including the number of the parameters and data samples used in the aforementioned baseline work, all the methods face the same training data inadequacy problem and almost no publications show the CT accuracy except our paper and the work in [9]. Although some results in the baseline work shows excellency in ST, the CT results in our work and [9] provide an estimation closer to how a consumer electronic product such as a tablet react to its new user in real-world scenarios. The results given in our our work also show that it is difficult for the models to generalize to data samples from the volunteers beyond the training set. This paper conducted the CT experiment on data collected by more different volunteers and verified that the fine-tuning method can better generalize the dataset to more people’s gestures than the proposed model in [9]. The second advantage of our work is that enriching a small auxiliary dataset, such as small shots of labeled gestures to the training dataset, can help to alleviate the effect of private user gesture habits. As the same gesture feature varies considerably between different users, enriching with a few gesture shots from a specific user is crucial for making the model become more adaptable to new users.

## 5. Conclusions

In this paper, fine-tuning and evaluation of current state-of-the-art CNN models in a MR-HGR task was performed based on the limited small data samples. A parameterized representation of a TSV spectrogram was proposed as a compact data modality of the time-evolving hand gesture features as the model input. The models used included ResNet with 18, 50, and 152 layers; DenseNet with 121, 161, and 169 layers; MobileNet V2; and ShuffleNet V2. Compared to the baseline models and previous work, this paper contributed in the cross-testing and achieved an average recognition accuracy of 93.47%. More importantly, this paper investigated the effect of personal gesture habits on the recognition accuracy degradation and propose the solution by enriching the dataset to improve the model’s adaption to the new gestures never learned. The work in this paper can be considered as a support to enhance the feasibility of MR-HGR in further applications.

## Figures and Tables

**Figure 1 sensors-21-00259-f001:**
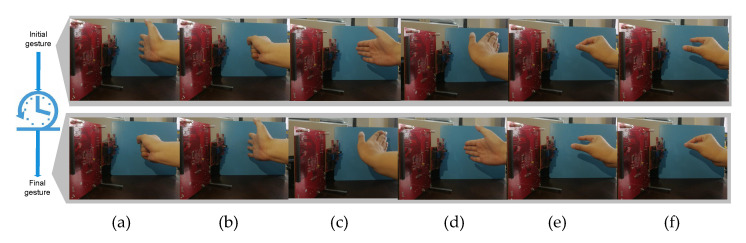
Visual images of six different hand gestures in front of millimeter-wave radar. Each gesture dynamically moves from the initial state to the final state with the shortest route: (**a**) page-close, (**b**) page-open, (**c**) page-up, (**d**) page-down, (**e**) page-zoom-in, and (**f**) page-zoom-out.

**Figure 2 sensors-21-00259-f002:**
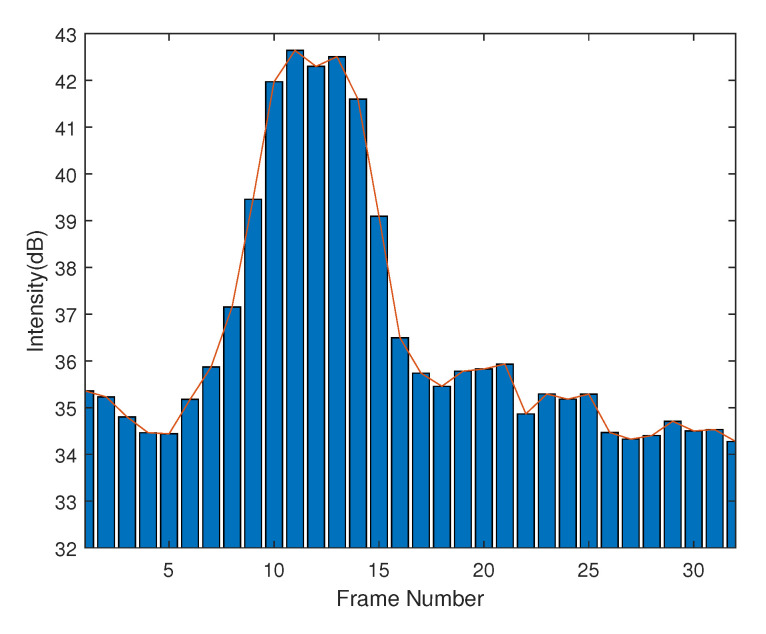
Entire energy variation for a gesture.

**Figure 3 sensors-21-00259-f003:**
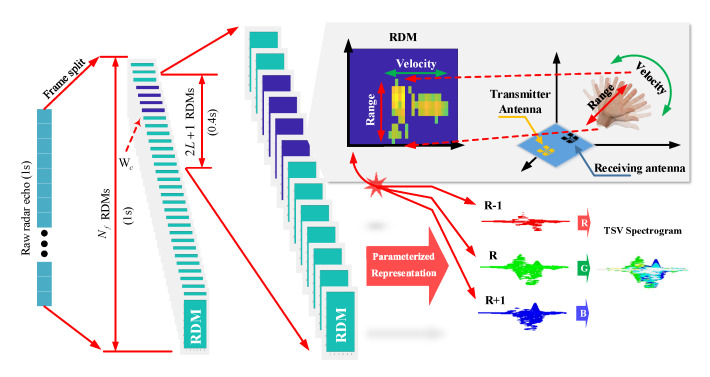
Temporal space-velocity (TSV) spectrograms of dynamic hand gestures. A demonstration of extracting the relevant 2L+1=13 of $N_{f}=32$ frames from a 1 second length of radar echo signal and determining the $W_{c}$. The actual range is 18.6 cm to 60.45 cm, which is the set gesture range. We selected the first 6 frames and the last 6 frames of **$W_{c}$** and a total of 13 frames with duration time 0.4 seconds to describe the gesture. If $W_{c}<L$ or $W_{c}>N_{f}-L$, we select 2L+1 frames from two adjacent measurements.

**Figure 4 sensors-21-00259-f004:**
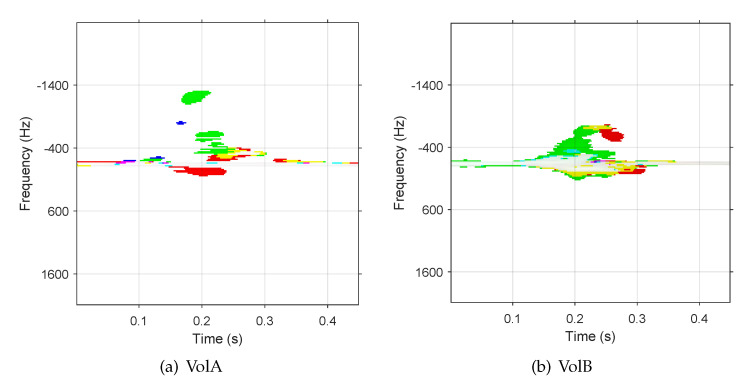
TSV spectrograms of ”Page-zoom-out” gestures performed by two volunteers. The (**a**) is done by VolA and the (**b**) is done by VolB.

**Figure 5 sensors-21-00259-f005:**
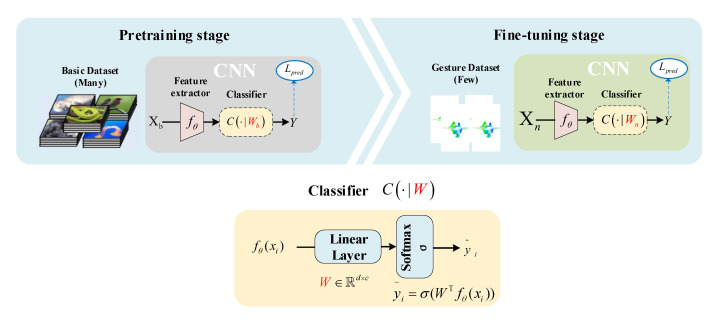
Fine-tuning method. A typical convolution neural network (CNN) model applied by a number of models is shown in this figure which usually consists of a signature extractor $f_{\theta }$ and a classifier C(·|W). In this paper, the basic dataset $X_{b}$ in pretraining stage is ImageNet [24], and gesture dataset $X_{n}$ is our local dataset described in Section 2.1. We denote the dimension of the encoded signature as *d* and the number of output classes as *c*. For all the classifier weights, $W_{b}$ and $W_{n}$, we have $W\in R^{d\times c}$.

**Figure 6 sensors-21-00259-f006:**
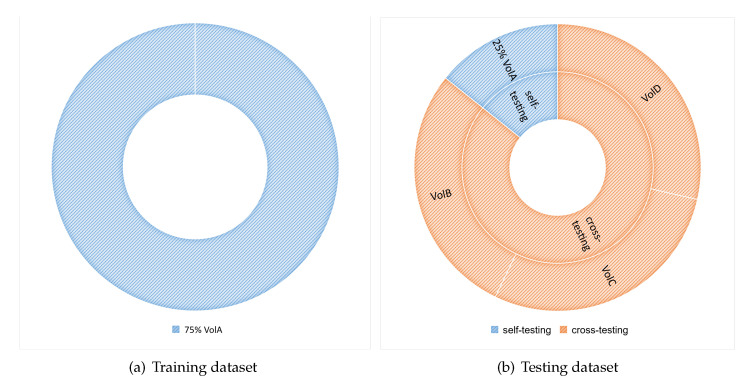
Training and testing dataset configuration of the baseline and fine-tuned experiment: (**a**) Training dataset: a total number of 900 training samples: 150 × 6 = 900 samples from VolA. (**b**) 300 samples from VolA as the ST testing dataset and a total number of 2304 samples from VolB, VolC, and VolD as the CT testing dataset.

**Figure 7 sensors-21-00259-f007:**
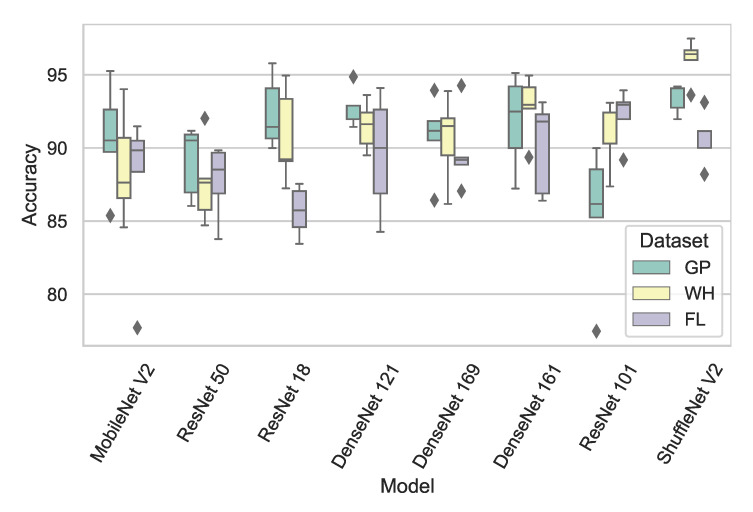
The CT box-plot regarding to each model after 5 folds with the radar data samples from three tested volunteers: VolB, VolC, and VolD.

**Figure 8 sensors-21-00259-f008:**
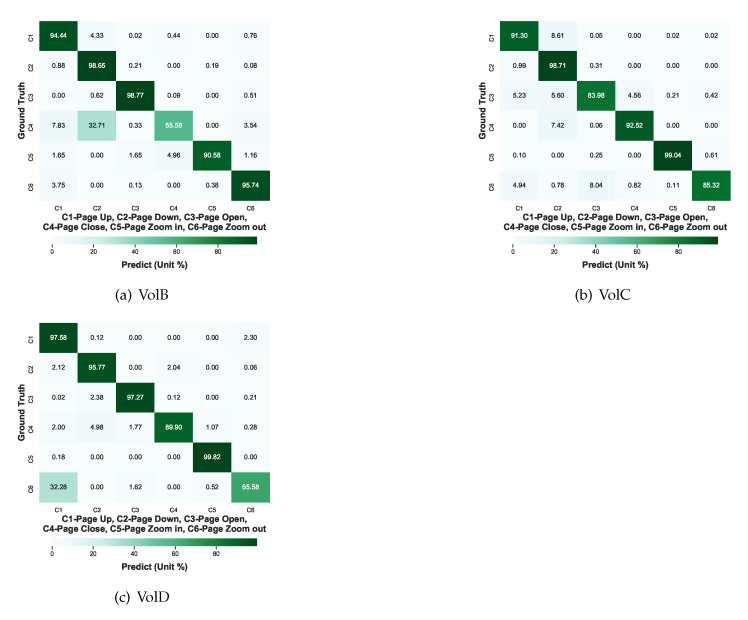
The CT confusion matrices of all the hand gestures after 5 folds regarding to each volunteer: (**a**) VolB, (**b**) VolC, and (**c**) VolD.

**Figure 9 sensors-21-00259-f009:**
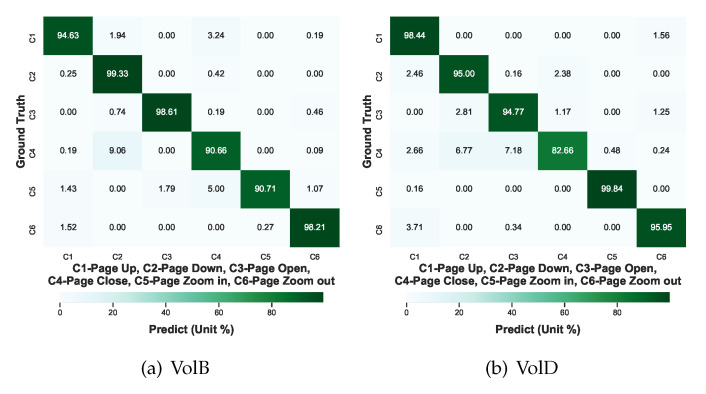
The CT confusion matrices after enriching regarding to each volunteer: (**a**) VolB and (**b**) VolD.

**Table 1 sensors-21-00259-t001:** Dataset configuration and volunteer information. Capital letters of different volunteer names are used as each dataset names, and Capital letter **M** means male and **F** means female.

**Volunteer** **Info**	**Name**	VolA	VolB	VolC	VolD
	**Age**	22	21	21	23
	**Height/cm**	170	165	178	179
	**Gender**	**M**	**F**	**M**	**M**
**Sample number** **per gesture category**	**Page-up**	200	128	128	128
	**Page-down**	200	128	128	128
	**Page-open**	200	128	128	128
	**Page-close**	200	128	128	128
	**Page-zoom-in**	200	128	128	128
	**Page-zoom-out**	200	128	128	128

**Table 2 sensors-21-00259-t002:** Cross-testing (CT) accuracy and self-testing (ST) accuracy in baseline experiment and fine-tuning experiment. The cross-validation accuracy is CT accuracy from Fold 1 to Fold 5.

Model		Baseline		Fine-Tuning						
**Name**	**Param**	**CT**	**ST**	**Fold 1**	**Fold 2**	**Fold 3**	**Fold 4**	**Fold 5**	**CT**	**ST**
ResNet 18	11.69	83.88	98.67	94.67	90.00	92.27	91.75	90.15	91.77	100.00
ResNet 50	25.55	72.18	98.67	84.91	89.25	87.79	89.39	89.58	88.18	99.83
ResNet 101	50.00	62.85	99.33	89.86	89.86	81.05	89.11	90.52	88.08	100.00
DenseNet 121	7.89	79.54	100.00	91.04	92.08	91.70	92.69	89.77	91.46	99.58
DenseNet 169	14.15	78.50	97.00	91.94	87.93	88.45	91.23	92.31	90.37	100.00
DenseNet 161	28.68	82.37	99.67	93.68	92.36	87.74	92.60	91.98	91.67	100.00
MobileNet V2	3.51	73.88	100.00	91.51	91.42	87.79	85.24	86.75	88.54	99.83
ShuffleNet V2	1.37	85.34	99.33	94.48	93.16	93.92	92.98	92.83	93.47	100.00

**Table 3 sensors-21-00259-t003:** Performance comparison with other baseline methods including volunteer numbers, gesture types, testing accuracy, parameter size, and example number.

Classifier	Volunteer Number	Gesture Types	Self-Testing Accuracy	Cross-Testing Accuracy	Params (M)	Training Examples	Testing Examples
CNN + LSTM [4]	10	11	87	-	689	1375	1375
CNN + LSTM [6]	-	7	91	-	<1	840	1960
CNN+LSTM+CTC [7]	4	8	96	-	6	3200	400
3 layer DCNN [8]	-	8	85.6	-	<1	450	50
3 layer DCNN [9]	2	14	92.9	48.36	<1	1792	448
u-DeepHand [10]	4	8	95	-	29.2	2000	500
CNN [19]	-	3	96.6	-	<<1	60	27
LeNet [5]	-	9	94.4	-	1.6	360	40
Ours(ResNet 18)	4	6	100.00	91.77	11.69	960	2124
Ours(ShuffleNet V2)	4	6	100.00	93.47	1.37	960	2124

## Data Availability

Publicly available datasets were analyzed in this study. This data can be found here: https://github.com/pantheon5100/MRHGR-dataset.

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
