# Peer review of "On the Effect of Training Convolution Neural Network for Millimeter-Wave Radar-Based Hand Gesture Recognition"

_sensors, 2021, doi:10.3390/s21010259_

Round 1

Reviewer 1 Report

Hand gesture recognition is a hot research topic and is widely used in various application scenarios. This paper proposes a millimeter-wave radar based hand gesture recognition (MR-HGR) method. Compared with other existing methods, the MR-HGR method is contactless and does not require to wear any device/sensor. The proposed method addresses two main challenges: 1) to deal with a small size of training samples, it transfers CNN models in computer vision to MR-HGR by fine-tuning these models; 2) to adapt the radar sensing data to the CNN models, a parameterized representation of dynamic hand gestures is proposed by using temporal space-velocity (TSV) spectrograms to represent different gestures. Experiments are conducted and multiple models are evaluated to demonstrate the performance of the proposed method. The experimental results are very promising.
The research topic of this paper is very interesting. The paper is well written and organized.

There are a few comments:
1. The experiment evaluation only includes six gestures. If possible, it would be better to include more gestures to fully demonstrate the performance of the proposed method.
2. In section 2.1, the authors indicate that "Nine volunteers proficient in using a tablet were invited to collect a total of 3324 gesture examples as the local datasets, named as shown in Table 1." It looks that there are 3624 samples in Table 1.
3. In Table 1, one user collects 200 samples for each gesture, three users collect 128 samples for each gesture, and five users collect only 4 samples for each gesture. This data set is extremely unbalanced for different users. This may impact the performances of the ST test and, especially, the CT test in the experimental evaluation part. For example, in Figure 6, the basic training dataset contains 900 samples from user GY but only 120 samples from each of the other five users. A more balanced dataset for different users would be better.
4. It may be not appropriate to disclose volunteer names in any publication, even only showing the capital letters of their names.

Reviewer 2 Report

In this manuscript, the authors present a method for dynamic hand gesture recognition based on millimeter-wave radar measurements and deep convolutional neural networks (CNN). They propose a parametrized, image-like representation of the data, suitable to be passed to the CNN network, and pay special attention to the network training strategy. They test their method using six common gestures performed by nine users and several known networks tuned by the transfer learning technique. Here are my comments and question:

  1. The description of the manuscript structure, given at the end of section 1, does not correspond to the real content.
  2. In the caption of Fig. 1, the letters (a), ..., (f) are not present in the figure.
  3. I guess that the numbers given in Tab. 1 mean particular gestures count. It is not so clear from the table caption and description in the text. I am surprised by the overrepresentation of GY's gestures and the small number of performances for ZK, LT, JC, LG, DL. Do not you think that the highly unbalanced data is not representative and may affect the recognition results?
  4. I do not understand the last part of the parametrized representation description in which you explain how you transform the selected sequence of RDMs into an RGB-like image (#164-166). Could you please explain it more precisely? 
  5. I do not feel qualified to judge the English language and style, but I have an impression that it should be polished and some spelling mistakes should be corrected, for example, 'spectromgrams' (#110).

Round 2

Reviewer 2 Report

All my comments and questions from the previous version have been taken into account. I believe the manuscript has been improved.

This manuscript is a resubmission of an earlier submission. The following is a list of the peer review reports and author responses from that submission.

Round 1

Reviewer 1 Report

The manuscript entitled “On the effect of training convolution neural network for millimeter-wave based hand gesture recognition”, proposed millimeter-wave based hand gesture recognition system, and compared the performance with ResNet, DenseNet, light-weight, MobileNet v2, and ShuffleNet v2. This manuscript is well written and happy to read. It has a good scientific contribution.

However, in the dataset section, the authors mentioned only 9 volunteers. As the gesture, posture, and style varies person to person, it is suggested to mention the age range and height.

Reviewer 2 Report

This manuscript presents a evaluation of CNN models for millimeter-wave based hand gesture recognition. The accuracy and computation time of each model was compared. Currently there are areas that require more details and the authors are recommended to address the following comments.

1. The motivation of this study is not clear. Also, some literature studies on other ges
   a) Why focus on millimeter-wave radar sensor for gesture recognition? Any advantages or disadvantages over existing sensors for gesture recognition? 
   b) Similar to the previous question, why focus on CNN? there are many other successful classification models for gesture recognition. The authors are advised to review existing research in more details.
   c) The manuscript lacks literature review on CNN for gesture recognition.

2. There are only two volunteers in this study which is considered very small. The authors are recommended to increase the number of volunteers and provide statistical analysis.

3. How are the experiment performed. Are the gestures performed in sequence (consecutively) or separately? The authors could provide an illustration for better understanding?

4. The method of fine-tuning the CNN is not clear. Are there any criteria used? Also, how do you guarantee that the model will not overfit?

5. The authors emphasize the importance of light weight networks for consumer electronics. if this is the case, how could millimeter wave radar benefit the consumer? Since this type of sensor are usually fixed at a certain location, it requires the user to be near the sensor. In what kind of use-case can the millimeter wave technology be used?

6. There are no comparison with other gesture recognition studies. There are other methods that are lightweight and produce satisfactory accuracy.

Reviewer 3 Report

General comments

In general, the paper makes sense. There is limited originality, but the approach is interesting, especially how the data is pre-processed and fed into the CNNs as images.  The fine-tuning approach based on a very limited data set, deserves to be described more clearly and evaluated properly. The main missing point in the Results section is the significance testing of the differences in accuracy between the best performing methods and the others.  

In HCI and interactive applications, the latency is crucial and hence I would at least expect in the discussion some more comments on that. A comparison of the latency values for the trained networks is needed in a table.

Now  all experiments are carried out on the same set of 9 volunteers, will personalised training be necessary in field applications?

Detailed comments are given below. Besides the numerous  phrasing problems, I have indicated the major comments in Bold.

Abstract

millimeter-wave based hand gesture recognition (MR-HGR) -- > millimeter-wave radar based hand gesture recognition (MR-HGR)

light-weight one, MobileNet V2 and … à light-weight MobileNet V2 and …

had shown -- > has shown up

Change phrase structure, e.g. : “Moreover, DenseNet 121 was 13,3% smaller in size and required  only 23,9 % of the inference time compared to ResNet 150.”

lightweight model like ... -- > lightweight model …

Section 1: Introduction

effective tool to represent learning method … -- > effective learning method …

was firstly presented -- > is firstly presented

we proposed -- > we propose

we conducted experiments  -- > we describe  the experiments 

,we gave -- > ,we give

Section 2.1

There hand gestures -- > These hand gestures

Caption Table 1: volunteer names as each dataset names -- > volunteer names are used as data set names

at R with  -- > at range R (?)

according to 3. -- > according to [3].

abduction/abduction -- > abduction/adduction

help model to -- > help the model to

The caption of Figure 3 needs some clarification: I did not find where nc is defined.

Section 2.1.2:

Concerning  generalization of the model to different people: The last phrase of this section is not fully clear to me. What is the little trick? This seems to me simply an ad hoc choice for splitting the data in training, validation and test data. Explain why you have chosen this particular split of data?

Figure 4: I do not see that volunteer 2 takes less time to complete the gesture. I also do not see that the volunteer 2 velocity is faster. Maybe the explanation for that can be taken up in the caption?

Section  2.2.1:

In later 2016 -- > Later in 2016

Section 2.2.4

upgrade version -- > upgraded version

“Concat operating concat all” It is not clear what you mean by this expression. Is it a specific setting for Shufflenet?

Section 2.3

The version that I found of Reference [25] has been written in Chinese language. It would hence be helpful if a short explanation of the transfer learning procedures in [25]. would be described here.

Xb and Xn should be defined in the caption of Figures 5

I am confused by the text. What is the pre-training and what is the training. The pre-training is already done based on ImageNet data? That is not the training stage depicted in Figure 5?

Somewhere in the paper you should mention which part of ImageNet was used  in the pre- training stage. Also a bibliographic reference would be appropriate when ImageNet is mentioned for the first time

Training examples in the computer vision – Do  you mean ImageNet examples?

Where are c and d further used in the paper?

Section 2.3 seems crucial to explain what you did exactly and should absolutely be clarified.

Section 3.2:

 same evaluation criterias -- > same evaluation criteria

epochs -- > epochs

 “checking of degradation problem suffered model” what do you exactly mean by that?

I agree that you do not go into data augmentation, but I fail to understand the reasons  for that, mentioned in the last phrase of the section.

Table 2 is ambiguous: you mention “training time per epoch” and later you state that TIME is coupled to one iteration so time should be multiplied by 100 to get the training time per epoch?

What do you mean by “which has more layers than it did”

Section 4 Discussion:

Is it feasible to add some accuracy data, the data set used and the number of gestures, … for the references that you mention, e.g. [1], [2], [4], …. [14] as a baseline for the qualitative performance comparison with your accuracies. A table could be added either in the introduction or in the discussion. A short discussion on this could then be added in Section 4.

You mention HCI for mobile applications in the introduction. I wonder what the latency for the trained networks is, to do a correct hand gesture recognition?

When the accuracies are compared, the results of a significance test has to be added for the difference between the best performing ones with respect to the others.

Round 2

Reviewer 2 Report

The reviewer would like to thank the authors for their efforts and time put into revising the manuscript. The manuscript has been greatly enhanced but the following key points need to be further clarified.

  1. In the introduction, the authors mentioned that this manuscript is intended to answer

    "(1) whether the fine-tuned network can work appropriately with the MR-HGR 55 dataset?" 

    What kind of challenge or issue are involve or expected in fine-tuning the network? It is usually expected that the fined-tuned network will perform better than the initial model. As such, the necessity/significance to address this question is unclear and should be elaborated in the introduction.

  2. Regarding the second objective of this paper

    "(2) what is the result of the evaluation under the criteria of computing time and test accuracy?",

    more discussion on how the classifier for a given task could be selected based on the accuracy/computation time tradeoff, should be highlighted. The current state of the manuscript mainly describes the accuracy and model complexity.

  3. Please provide more in depth analysis of classification result of each volunteer. For example
      a) What are the classification results of each volunteer?
      b) How do individual results change with model size/type classifier?
      c) Are there any degrading of classification results for a particular individual even though the model is more complex. if so why?

  4. In the author's response, the authors claim

    "There are many existing researches focus on vision based hand gesture recognition, which collects data with a lot of background information that is often related to the privacy of the user, and such methods usually have high requirements for lighting conditions. Since the detectable distance of millimeter waves is very limited, this protects the privacy of users to some extent, and millimeter waves are less disturbed by the environment can work in many scenarios such as dark night, foggy weather. And the use of millimeter wave radar for gesture recognition is contactless, while the reduction of contact manipulation in public places such as elevator buttons can reduce the spread of viruses. At the same time the use of millimetre wave radar does not suffer from illumination problems"

    The reviewer strongly recommends the author summarizes the pros/cons using a table for better understanding of the motivation and should be included it into the manuscript. The table should also include comparison with other wearable sensors for gesture recognition such as electromyogram (EMG).

    Also, the reviewer is not convinced on how the superiority of millimeter wave radar for commercial gesture recognition products. Since the main disadvantage of this method is that the sensors are usually fixed, which means that in order to do gesture recognition, the users are constrained near the sensor. Whereas, wearable sensors can perform gesture recognition anywhere as long as the user wears them.

Reviewer 3 Report

The authors have replied to my remarks, content-wise. They adapted the paper accordingly. By doing so there are again some new phrases that make no sense grammatically. I made a list of the major cases below, but a serious editing of the English language is still required: 

Line 22 watch[1]. -- > watch [1].

Line 28 In this case, numerous state-of-the-art methods in this field -- > Suggestion: “ In this case, numerous state-of-the-art methods for hand gesture recognition  ##What is “field”?##)

Line 36 target gestured in front -- > target gesture in front ##?##

Line 89 a Off-the-shelf 7 -- > an Off-the-shelf 7

Lines 104 and 109  formality. I think you want to say “form”  ##?##

Line 119  respectively represents -- > respectively represent

Caption of Figure 4    frame, select -- > frame, we select

Lines 164-168 I would suggest changing the phrase: The difference can be interpreted by the intrinsic seperating velocities of the thumb and index finger when the two volunteers demonstrate the page-zoom-out gesture individually and also affacted by their different palm sizes and finger lengths. -- > This can be interpreted  from the different behavior of the two individuals when performing the page-zoom-out gesture, more specifically from the two separate velocities of the thumb and  the index finger in the page-zoom out gesture. The TSV  spectrogram  is also affected by the different palm size and finger lengths of the two individuals.

Line 171 FL,GP and WH is testing data sets totaling … -- > FL,GP and WH are used for testing, totaling …

Caption Figure 4 frame, select -- > frame, we select

Lines 209-210

… which is switched from add to concat operating. -- > … which is switched from addition  to concatenation operation.

 These concat operatings can preserve … -- > These concat operations can preserve …

Line 221 usually refer -- > usually refers

Line 262

We fine-tuned all the models with a relatively small learning rate 0.001 and a batch size of 20 to avoid GPU memory outage, and the number of epochs, 50 and 100, were chosen for checking whether the model suffered from degradation problem and the fastest converging model. -- > We fine-tuned all the models with a relatively small learning 2 rate 0.001 and a batch size of 20 to avoid GPU memory outage. The number of epochs, 50 and 100, were chosen for checking whether the model suffered from degradation problem and the fastest converging model.  ## the last phrase is grammatically incorrect and hence unclear##

Line 264 further process -- > further processing

Caption Table 2  the duration of model training time in per epoch -- > the model training time per epoch

Line 274-276: The added/modified phrase is really cryptic. Please express this clearly,

Line 333 literature are -- > publication is

Line 338 and following lines (also Table 3) “cross-testing”. What exactly do you mean by that? Am I right that cross-testing is testing on the gestures performed by another person than the one for which the network was trained and self-testing is testing on gestures not used in the training but coming from the same person than the one used for training? Maybe you  could explain that more clearly in the text that was added if this my assumption is correct, otherwise I do not understand what you want to say.

Line 342 only the successive operations of some basic gestures -- > only the execution of some basic gestures ##is that what you want to say?##
